# A Systematic Review of Financial Literacy Research in Latin America and The Caribbean

**Silvia Mariela Méndez Prado** [1,*], **Marlon José Zambrano Franco** [1], **Susana Gabriela Zambrano Zapata** [1], **Katherine Malena Chiluiza García** [2], **Patricia Everaert** [3] **and Martin Valcke** [4]

1   Facultad de Ciencias Sociales y Humanísticas, Escuela Superior Politécnica del Litoral, ESPOL, Campus Gustavo Galindo Km. 30.5 Vía Perimetral, P.O. Box 09-01-5863, Guayaquil 090902, Ecuador; marjzamb@espol.edu.ec (M.J.Z.F.); suszambr@espol.edu.ec (S.G.Z.Z.)
2   Information Technologies Center, Escuela Superior Politécnica del Litoral, ESPOL, Campus Gustavo Galindo Km. 30.5 Vía Perimetral, P.O. Box 09-01-5863, Guayaquil 090902, Ecuador; kchilui@espol.edu.ec
3   Faculty of Economics and Business Administration, Ghent University, EB22, Sint-Pietersplein 7, 9000 Ghent, Belgium; patricia.everaert@ugent.be
4   Department of Educational Studies, Faculty of Psychology and Educational Sciences, Ghent University, Henri Dunantlaan 2, 9000 Ghent, Belgium; martin.valcke@ugent.be
*   Correspondence: smendez@espol.edu.ec

**Abstract:** Several well-known studies have remarked on the low financial literacy (FL) levels in Latin America and the Caribbean (LAC), which represent a problem in an economic context of change and uncertainty. This fact gives us the opportunity to evaluate the current state of literature related to FL in the region. The main list of identified keywords allowed the PRISMA methodology to guide the systematic literature review and analysis procedure. During 2016–2022, the FL search yielded around 4500 FL manuscripts worldwide, but only 65 articles were related to the scope of our analysis (which involved looking at LAC countries). Being the first review from an LAC country about all LAC countries, the findings highlight a lack of FL research focus on regional needs, gender gaps affecting women, and conceptual frameworks used to develop efficient educational program interventions. Most studies in this review build on the OECD definition of FL, but the financial attitude dimension often seems to be omitted from the analyses. These findings open the discussion about efficient policy design concerning FL development in LAC.

**Keywords:** financial literacy; LAC countries; PRISMA; systematic review

## 1. Introduction

Recent decades have demonstrated that we are living at a time of fragility and constant uncertainty due to consecutive economic and political system changes [1]. These facts stress the increasing importance of raising financial literacy (FL) levels in the population to ensure that individuals achieve adequate financial wellbeing [1]. Vieira et al. (2018) described FL as knowledge, attitudes, and behaviors that play a key role in enabling people to make assertive decisions regarding their financial wellness [2]. FL is crucial in facing challenging situations in a particular country and setting. It allows individuals to control their involvement with financial products in the market and to manage their finances [2]. With the expansion of financial services and financial–technological banking systems, the need for more in-depth financial knowledge (FK) has attracted the interest of scientists in developing research, experiments, and reviews about FL [3].

Researchers have also explored the impact of FL on key economic variables and related wellbeing levels by studying, for instance, individual wealth accumulation, debt, risk tolerance, and retirement preparedness. The relationship between FL and certain kinds of economic and financial behavior has been well documented [4,5].

This is reflected in the growing number of FL publications. Researchers try to map financial knowledge (FK) and how individuals act or what attitudes they hold toward

various financial situations that arise daily. Studies tend to focus on access to banking products, FK, and bank loans, and they link these to individual demographic variables influencing potential literacy gaps [1]. However, the literature remains limited and suggests that FL in most countries is low, even in countries with major stock markets, such as the United States or the United Kingdom [6]. Research shows that developing countries have lower financial education and literacy rates [7]. This is especially true in the context of Latin American and Caribbean (LAC) countries; however, a shortage of studies analyzing the state of FL in these countries exists [8].

The current study tried to fill this gap by describing the setup and results of a literature review about the current FL situation in LAC. The following objectives directed the study: to describe the most important contributions that have been made in recent years in LAC to determine the progress that has been made in FL in this region; to find the approaches that FL research has taken in recent years to delineate fields of interest; and to provide a summary of articles published in LAC as a guide for future research.

This paper is divided into six sections. Section 2 explains the conceptual framework. Section 3 describes the selection process for the analyzed information. Section 4 discusses the findings of the selected papers. Section 5 presents a discussion regarding the revisions that were made to this study. Finally, Section 6 highlights the main conclusions.

## 2. Conceptual Framework

FL is a topic that has gained relevance in recent years. Researchers have developed several concepts to describe its meaning and implications. Therefore, to understand the content of this research, it is essential to explain this core concept. One of the best-known definitions proposed by Lusardi et al. (2010) describes FL as general knowledge about financial concepts, such as compound interest and nominal and genuine interest, and the ability to make effective decisions regarding personal finance, savings, investments, and other related topics [9]. Similarly, the Organization for Economic Co-operation and Development (OECD) has described FL as "knowledge and understanding of financial concepts and risks. The skills, motivation, and confidence to apply such knowledge and understanding to make effective decisions across a range of financial contexts, to improve the financial wellbeing of individuals and society, and to enable participation in economic life" [10].

On the other hand, FL can be understood as a fundamental pillar for individuals to develop their knowledge and achieve financial stability [11]. Based on the abovementioned definitions, FL involves learning about financial issues. Financially literate individuals will develop skills and abilities that will lead them to control their finances better. In turn, this will allow them to make effective decisions that guarantee their individual wellbeing.

Warmath and Zimmerman (2019) defined "FL as one's capacity to make effective financial decisions, where "capacity" refers specifically to knowledge, skill, and self-efficacy [12]". Aydin and Akben (2019) stated that "the interrelationships between the three dimensions of FL as well as on how money attitudes and time preferences affect FL among the young [13]". Kamiya (2017) remarked that "the early definition of FL meant "financial knowledge", but the latest definition now includes or refers to consumers' financial behaviors, consumers' interactions with their social and economic environments. The effect of cognitive biases on consumers' financial behaviors [14]". Going further, Potrich et al. (2016) not only analyzed the interrelation of different FL dimensions but also found that the FL of a sample of university students was dependent on their financial behavior, financial knowledge, and financial attitude, with the latter having the greatest impact [11]. To summarize the available literature, FL seems to be dependent on three fundamental pillars: financial knowledge (FK), financial behavior (FB), and financial attitude (FA). These are also called FL dimensions.

Each dimension contributes to a person's ability to make decisions in pursuit of their financial wellbeing, and thus it is necessary to consider each of these dimensions to build their FL [10,15]. The FA dimension considers how the individual perceives and

judges financial issues, that is, the person's intention and evaluation regarding money and its use [16]. On the other hand, FB comprises an individual's skills and actions as they pertain to achieving short-term and long-term financial goals, which can be future acquisitions or unforeseen expenses [17]. Finally, the definition of FK considers assimilating and understanding information from economic or financial processes to make correct decisions on various topics such as financial planning, budgeting, lending, etc. [18]. Each aspect plays a vital role in individuals achieving high levels of FL.

As stated earlier, the present study tried to map the current FL situation in LAC. The systematic review aimed to provide a broader perspective of the state of the existing literature while focusing on the description of the available studies, the nature of FL definitions, and the qualification of the research.

### 3. Literature Review Methodology

The Preferred Reporting Items for Systematic reviews and Meta-Analyses (PRISMA) statement guided the present literature review [19]. The PRISMA approach suggests that authors explain the process of their research transparently and adequately [19]. Based on PRISMA, this study selected the most relevant research articles related to FL in LAC between 2016 and 2022 (23 January).

The Scopus database yielded 4465 manuscripts worldwide with "financial literacy" as a keyword. We noticed an exponential growth in the number of publications made from 2008 to 2022, evidencing the growing interest in FL research.

After the OECD promotion of FL in 2008, the first year of our analysis, 52 FL articles were published. Eight years later (2016), the number of papers published on the topic annually exceeded 300.

We delimited our work to the last five years (2016–2022), considering that the most recent lustrum accounted for more than 60% of the research published on the topic. In 2016, there were more than 300 global publications (333 to be exact). The increasing slope of the contribution curve reached its highest point in 2021, with a good rhythm and increasing further in the first month of 2022.

First, the keyword "financial literacy" was introduced to the Scopus platform's search engine, which provided a total of 4465 results. Second, the publication interval was narrowed to 2016–2022, resulting in 3008 entries. Subsequently, we focused on studies in LAC and considered entries from 30 countries [20]. LAC usually refers to thirty-three countries, but we chose to use thirty—omitting Barbados, the Bahamas, and Cuba, from which no FL data are available.

Following these steps left us with 97 entries. Finally, the search was limited to articles, systematic reviews, papers, and books, leaving 95 results. Detailed analysis discovered that further 30 did not fit the FL field or did not match the regional context. This process left 65 studies to be included in the review. These publications were further classified by article type while focusing on scales to measure FL, interventions to study the impact of instructional activities, systematic reviews with critical evaluations of related studies, and longitudinal studies focusing on the evolution of FL variables over time. Figure 1 shows the inclusion/exclusion process:

A detailed analysis of the 65 articles resulted in a structured table scanning all selected papers for 14 proposed key variables based on our experience of what is relevant and common in FL studies (Table 1). We offer a transparent and comparable scheme of analysis to accomplish the third objective of this study— "to provide an LAC studies summary to guide future research". As depicted in Figure 2, the variables are journal, country, type of article, author, and year of publication. Other variables helped scrutinize the nature of the study such as FL definition, FL dimensions, FL scales, validity/reliability of instruments, contribution statement of the study, age/scope of the sample, sample size, and follow-up information (results, implications). Table 1 offers a detailed overview of the analytical process.

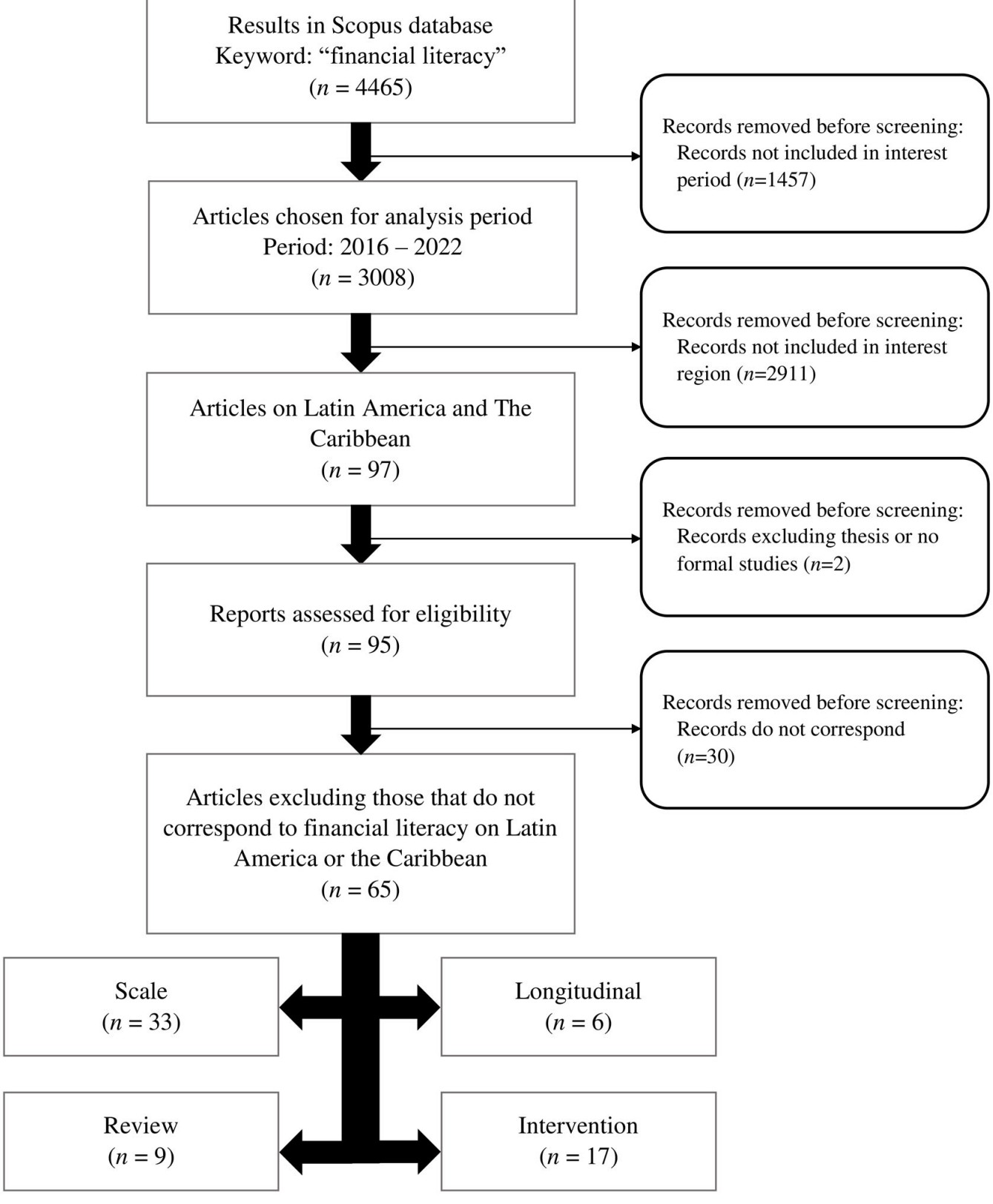

**Figure 1.** Literature selection process following the PRISMA methodology.

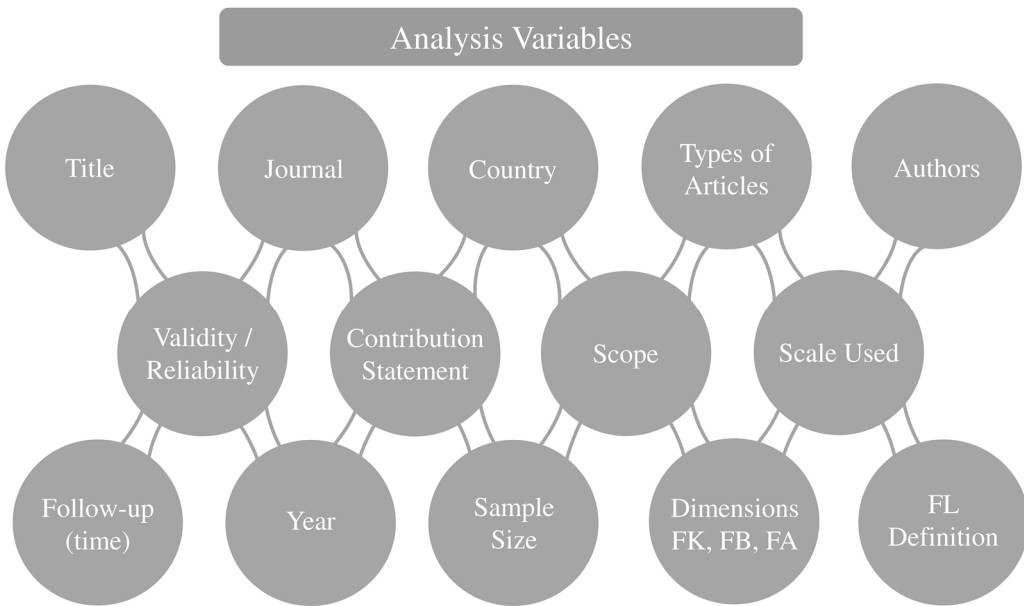

**Figure 2.** Variables for processing FL articles during our research.

PRISMA and the proposed scheme allowed us to first describe the most important contributions made in recent years in LAC. Second, it allowed us to determine the progress of this region in FL to find the approaches that FL research has taken in recent years to delineate fields of interest. The PRISMA process allowed us to formulate a method to find the 65 suitable manuscripts related to LAC (See supplementary materials). Table 1 shows a structured output that acted as the third research objective and allowed us to make the comparison needed to accomplish objectives 1 and 2.

Based on the above and to the best of our knowledge, the results section shows the studies' analysis structure to meet the first two research objectives. The proposed subsections in the Results section are publications over time, publications by country, FL definitions in LAC studies, type of FL studies (review, scale, intervention, longitudinal), and the endogeneity bias in LAC studies.

## 4. Results

### 4.1. FL Publications over Time

Despite producing increasing global and historical FL results, Figure 3 reflects the low output of articles related to FL in the LAC region during the last six years, which increased along with the worldwide trend in 2021. This increase could be explained by the higher accessibility of banking products and the FL background that navigating these necessitates [21]. The 2021 studies focus on investigations involving younger generations, primarily students, with stronger perceptions and attitudes toward financial issues but lower financial skill and usage levels. The latter could be associated with gender gaps [22]. For example, the technological gender gap intensifies when young adults have low levels of FL [23,24], with women lacking access to ICTs (information and communication technologies). The gender gap reflects—as has been reported globally—that these women seem to master FK to a lesser extent [25].

Individual behavior is a marked trend in the 2021 studies, as they provide critical information about what drives people to make decisions that influence their financial wellbeing. The study of Hernández-Mejia et al. (2021) contributed with an analysis of the behavior of individuals who own a credit card and have knowledge about finance. They were able to determine that women have a low level of financial culture and that those who own a card are more likely to pay their debts [26]. This study was carried out based on the increasing use of financial instruments such as credit cards. On the other hand, the study of Ali et al. (2021) was based on individuals' access to financial products

and the socio-political and economic situation experienced by Latin American countries, which may cause greater indebtedness of users [27]. The authors analyzed the FA and FB of history and geography university students, focusing on rationality and centrality. The behavior of individuals shows the gaps that can be caused by inequality of gender, socioeconomic level, and technology or all in a single case, as in the study by Hernández Rivera and Rendón Rojas (2021) [24].

During recent years, some studies have linked FL with various fields such as mental health, technology, and gerontology. These contributed new and creative approaches to the literature. For example, in the study by Notargiacomo and Marin (2021), the contribution was based on the description of a program developed with artificial intelligence called STIMA, which seeks to collect financial information from experts to provide favorable options to students at the time of decision making. A quasi-experiment was conducted to validate the program where a student used the tutor-supervised program for four months [28].

Other studies did not focus on measuring knowledge or demonstrating how dispersed groups can be found in terms of FL but on providing solutions to various problems that individuals experience in the financial sphere. In the study by Genta Maragni et al. (2021), their contribution was not only about the financial field; it included research on mental health. The authors sought to provide the possibility to undertake and improve the finances of people with mental illnesses, providing a new focus on the financial inclusion of a vulnerable population often undervalued in the workplace [29]. Duch et al. (2021) focused on a vulnerable group, retirees who receive pensions from private and public systems, and the authors carried out a field experiment to improve the retirees' financial wellbeing [30]. García Mata (2021) analyzed the effect of retirement planning, taking as an essential axis the gender of individuals to measure intentions after having answered several questions belonging to the National Survey of Financial Inclusion based on a theoretical financial planning basis [31].

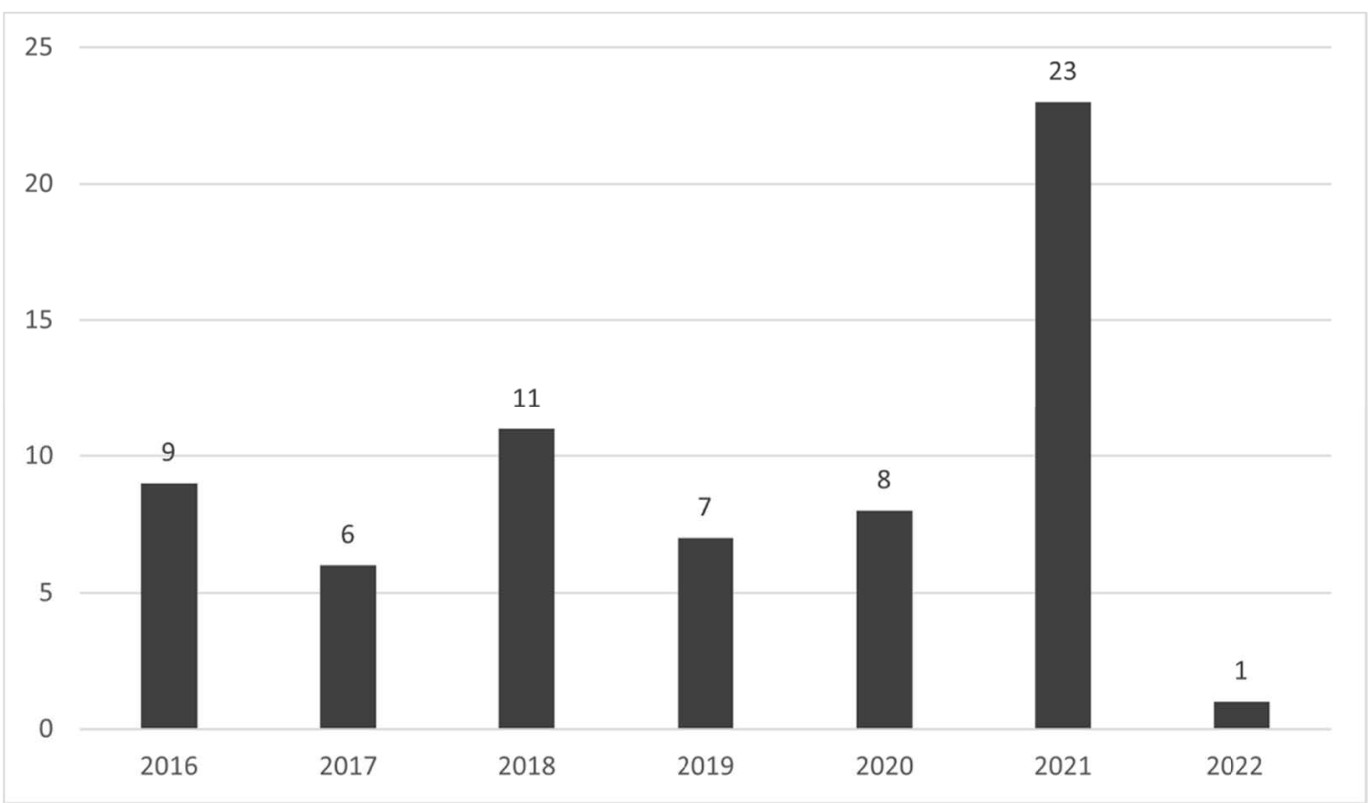

**Figure 3.** Selected FL articles published in Scopus from 2016 to 2022 (23 January).

### 4.2. FL Publications by Country

Regarding the country distribution of published articles (Figure 4), we found studies related to FL in only 8 of the 30 LAC countries. The highest number of studies were conducted in Brazil, with studies analyzing different aspects of FL and a focus on formal (business) education, again on gender differences. Being male results in higher levels of financial education [32,33]. A scale developed to measure the individual FL of Brazilians found that single women with low incomes and low financial education usually have low FL [18]. This article also highlighted Brazil's attempt to counteract the gender gap through women's financial support policies. The Bolsa Familia program, for example, was inefficient according to the findings of one study because giving women more money does not help them if they do not have an adequate level of FL to know how to manage it. That is why studies about financial education get a lot of attention in this country, focusing on personal finance courses [34] and designing innovative FL interventions, such as those based on video games, to impact FL in younger generations [35]. However, these studies consider the importance of adequately establishing financial education programs because FL is related to the individuals' characteristics [33]. Thus, diagnosing their profiles to develop financial solutions is crucial [36], for example, designing content focused on early levels of education to prepare children before they become adults or developing more specific courses that do not focus solely on theory but include practice examples to impact FA [17].

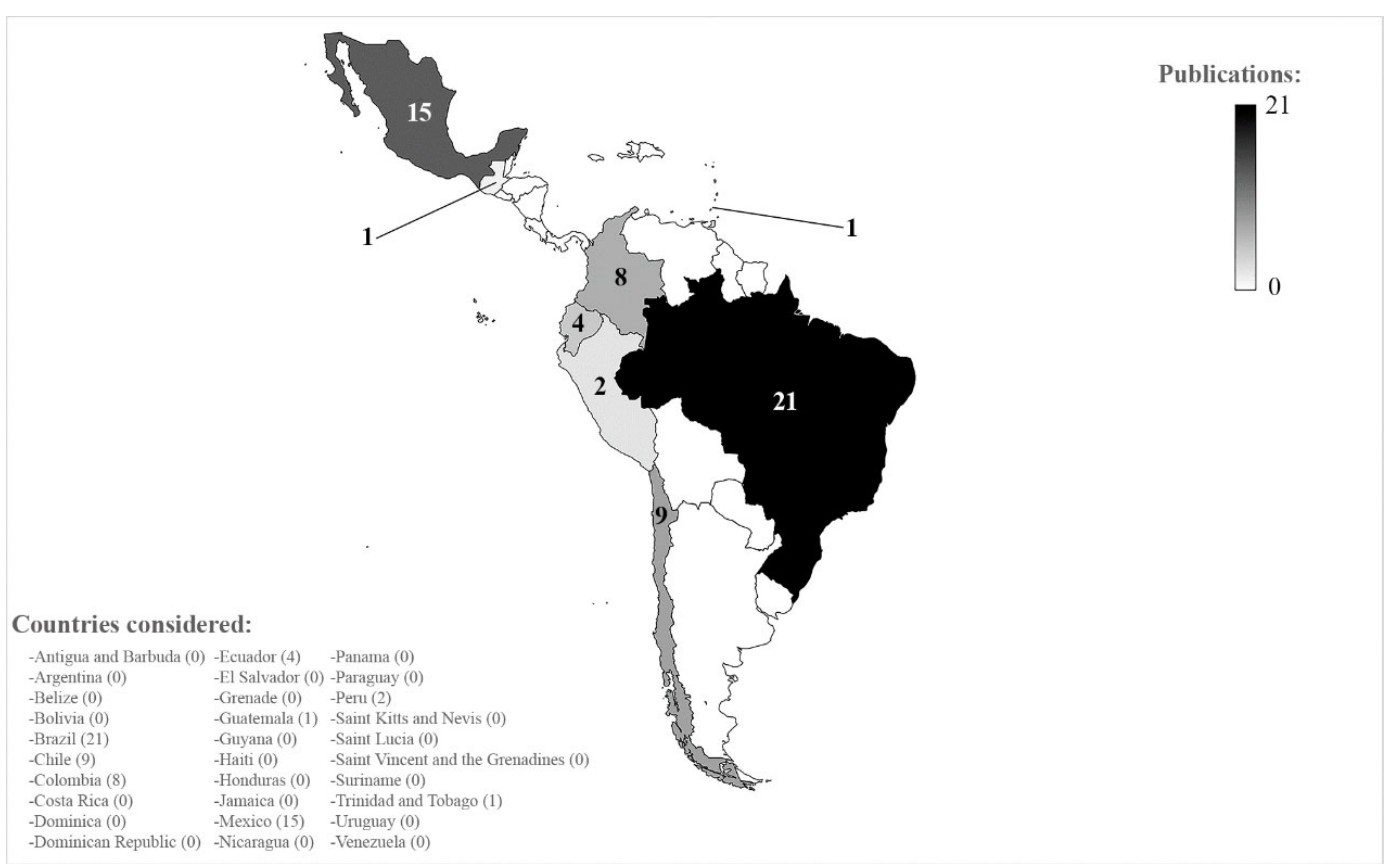

**Figure 4.** Articles registered in Scopus from 2016 to 2022 (23 January).

Furthermore, 15 articles were the result of research conducted in Mexico, with some of these studies involving high school and university student samples. The results point to low levels of FL without stating gender differences [24,37–42]. Studies set up in Chile (*n* = 9) involved more complex samples, with some focusing on micro-entrepreneurship, which exhibited positive FL outcomes that were reflected in decreasing debt levels [43]. Micro-entrepreneurial women, who have lower levels of FL, may therefore benefit more

from these interventions. Chilean studies also focused on pre-retirement samples [30] and subjects with loans or loan intentions within the next three months [44], concluding that rich information about retirement plans and loans positively affects the quality of decision making. The reason behind the selection of samples with complex characteristics in Chile comes partly from scandals that have existed around issues related to loans, such as the case of "La Polar" and "Cencosud", who took advantage of individual clients' financial assets [44].

Colombia and Mexico mainly present studies involving students [23,45,46] focusing on FL, knowledge, and skills, emphasizing the lack of financial development in students, and noting higher levels of FL in students with higher cognitive abilities. On the other hand, in Ecuador, the implementation of tools to study FL interventions stands out, such as video vignettes [47,48] that have a significantly positive effect on FL levels when the intervention is extended. As far as Guatemala is concerned, the only publication from the sample used a research design with demand-driven interventions focusing on inclusive health insurance in vulnerable populations [49]. Finally, the studies conducted in Peru and Trinidad and Tobago emphasize financial education components that contribute to FL and digital (financial) skills.

### 4.3. FL Definitions in LAC Studies

Authors' definitions of FL in LAC build mainly on the OECD's definition [50], which considers FL to comprise FK, FA, and FB, where each is required to make appropriate financial decisions that improve an individual's lifestyle.

The context in which each author considers the definition of FL is also highlighted. In most FL studies, the focus is on the knowledge and skills of individuals, working on strategies of individuals involved in small and medium-sized enterprises [51], the capitalization of bonds [52], and how individuals manage their pension fund portfolios [53]. The focus on the different FL dimensions helps to explain the low FL levels in the region [21,23,24,37–42].

According to Table 1, 44 of the 65 publications consider at least two of these dimensions in their FL definition, confirming the relevance of the conceptual framework across LAC studies. Fifty-nine (91%) publications consider FK in their definition, including knowledge of financial terms such as interest rates, inflation, purchasing power, and risk management. FB is tackled in 42 (72%) of the 65 publications, and 42% of the studies include FA as a critical research dimension. The lower focus on FA can be explained by the difficulty of measuring the opinions and judgments of individuals regarding finances, creating a risk of generating biases in FL measurements. This is related to authors having to use a proxy to measure FL given the limitations of the data [52–54]. However, the FA dimension has the same relevance as the other two dimensions, as is demonstrated when comparing the different approaches of the definition of FL [2] and the relation between the three dimensions [36].

The general research findings point to poor FK and FB in LAC. The results stress the need to strengthen the basic understanding of financial concepts and to build a financially responsible culture characterized mainly by rational decision making and adequate financial planning. Rational financial decision making is critical since the studies also remark the low self-control in money management and budget planning.

When comparing the situation in these countries with global development, the importance of FL in decision making is highlighted, as well as the need to consider all FL dimensions as pillars to achieve financial wellbeing [34,55–57]. Most authors consider at least one dimension of FL in the available research, thus making it possible to pull together the results of different studies [13,58–61]. Another shared feature is the emphasis on the impact of different levels of FL on managing personal finances in the short and long term [3,61–66].

### 4.4. Type of FL Studies

Analyzing the research presented in the FL studies helps us to understand the development of FL research and to classify it as focusing on the measurement scale, FL intervention, longitudinal studies, and review studies. Figure 1 and Table 1 point this out and document the number of articles of each type.

#### 4.4.1. Measurement Scales

As is presented in Figure 1, 33 articles reported ways to measure FL (50.77%). One of the most referenced scales is the FL-ABK questionnaire [67]. This scale is congruent with the above multidimensional FL definition with 40 questions that analyze the dimensions FA, FB, and FK.

Other studies build a scale to study FB—see the scale of Xiao and O'Neill (2018), the OECD (2013) scale, and the scale of Van Rooij et al. (2011). These scales measure FK based on questions tackling financial topics, calculating interest rates, and understanding the value of money, shares, mutual funds, and risk diversification [50,68,69].

Next, a considerable number of articles build on author-designed scales about financial perceptions [23,70], socioeconomic literacy [38,71], and a measure of FL in university students [23,45]. A particular study not only measured FL at one moment but generated a life cycle profile [72].

Many recent studies have provided contributions focusing on measuring FL and validating instruments known as scales. The study of Santoyo-Ledesma and Luna-Nemecio (2021) performed a descriptive experiment to validate a scale to measure the level of FL of the millennial generation [73]. In the study by Avendaño et al. (2021), a scale was also used to measure the level of FL, which collects the sociodemographic information, perceptions, and skills of individuals in various financial situations. It was possible to identify a low level of knowledge and a lack of programs that improve university students' competencies [23]. Scales can measure the level of FL in a general way and the dimensions' performance, depending on the purpose of the study. For example, in Paraboni and Da Costa's (2021) study, the three dimensions that make up the FL of university students were measured. Still, unlike the studies mentioned above, an intervention was carried out to measure their learning after a certain period [34]. Several studies in Latin America use scales to facilitate measurement. This fact potentiates a more significant number of articles adapting instruments, experiments, or quasi-experiments to increase the financial wellbeing of more and more people due to the greater access they have to financial products for which a higher level of FL is necessary.

#### 4.4.2. FL Interventions

Reports on interventions were found in 17 studies (26.15%). Interventions studied the effects of formal and business education on financial education [32], the impact of communication of banking stress test routines from the perspective of the consumer on FL [74], and the effect of highlighting an operational focus on cash flow in loan decisions [75], personal finance courses [34], and the use of video vignettes [47]. Meanwhile, one study focused on associations with cognitive abilities [46].

Some interventions measure the FL level with a proxy. For example, an analysis of the effect of individual choice on pension wealth considered that individuals who move away from the default pension fund portfolio are more financially literate [53]. Another example is capitalization bonds instead of a savings account. In the Brazilian context, the returns of these bonds (chance of winning a prize) are lower than the returns of a saving account [54]. The intervention studies reflect implicit models about the nature of the relationship between the interventions, various dependent covariates (e.g., gender), and mediating/moderating variables. However, it is striking that the studies do not present a clear theoretical framework to ground the research questions or hypotheses. There is no established way to guide people's FL [2,33,43]. Therefore, most authors who want to impact FL levels quickly base their intervention designs on definitions associated

with financial education [27,29,32,34,74]. All of them consider that FK without a proper theoretical approach ignores the need to improve behavior and attitudes of individuals.

### 4.4.3. Studies of FL Review Type

A review embraces a critical evaluation of earlier studies, which may or may not include an estimate of the effects [76]. Nine publications could be identified as FL review studies (13.85%). These reviews center on financial education in the Caribbean [77], access to bank credit for millennials [78], and financial innovation [79]. One meta-analysis stands out [33]. It sought to determine the background and consequences of FL. These kinds of reviews have either a global focus or a one-country focus, but none have studied the LAC region specifically. This is in keeping with the contribution of this work, which analyzes the evolution and principal axis of the FL studies in this region.

### 4.4.4. Longitudinal Studies

In total, six studies (9.23%) looked at variables within the same individuals over a more extended period. The studies focused on an extensive follow-up time window. One carried out an analysis of the relationship between financial education and household wealth in Bogotá over a 4-year period [80]. Another studied 66 years of discussion on FL in households in Mexico [81]. The shortest study looked at the effects of a video vignette intervention on participants' FL levels after one year [48]. The studies involved large samples [82] like Escudero and Ruíz's (2021) research looked at the influence of brokers' advice on people's willingness to pay, and small samples [83] like Téllez-León et al. (2019) looking for the main determinants of financial deepening. In general, the longitudinal studies look at later financial results—next to a focus on the intermediate changes in FL. These longitudinal studies become relevant when assessing the best way to impact FL on individuals. Financial education programs are more efficient in the long term [48]. This goes along with the considerations mentioned before, which highlight the adequacy of education programs to get better results.

### 4.5. The FL Endogeneity Bias in LAC Studies

Sekita et al. (2022) referred to the endogeneity bias in FL by explaining that wealthier individuals could, for instance, acquire higher FL through their higher exposure to risky financial assets [5].

The potential endogeneity of FL is addressed broadly by instrumental variable (IV) or generalized method of moments (GMM) techniques, according to Pesando (2018) [84].

Pesando (2018) grouped the IVs into three categories: (1) family background and financial knowledge of the peer or reference group, (2) information on past education and previous financial or math knowledge, and (3) instruments that exploit natural experiments or geographical variation in specified outcomes [84].

For example, in Japan, Watanapongvanich et al. (2021) controlled the endogeneity bias between FL and gambling behavior, using the education of respondents' fathers as an instrumental variable [85]. Noviarini et al. (2021) investigated the effect of FL on debt ownership, debt anxiety, and risk tolerance of older individuals in New Zealand. They found variate relationships by subsample cohort, and they remarked that the assumption of a simplified nature of the interrelation factors misleads relationship generalizations [4].

Looking at the endogeneity bias as a studied issue worldwide, the authors searched for FL plus endogeneity in the Scopus database. This revealed 33 published works dealing with FL endogeneity bias from 2016 to January 2022—45% of them from 2021. The increasing FL research production addressing the endogeneity bias is led by China, the United Kingdom, Japan, the United States, and Ghana. In the studied sample, Silva et al. (2021) from Brazil, a leading FL research country in the region, tested how consumers of banking services value stress tests performed by their banks [74] by applying IVs. This problem is not adequately recognized and registered as a common issue among LAC countries' FL research.

**Table 1.** Articles detailed by title, journal, country, type, the scale used, citation, year, the total number (#) of applied dimensions (financial knowledge (FA), financial behavior (FB), financial attitude (FK)) by using the cross symbol (X), financial literacy definition by the author, validity/reliability (v/r), contribution statement, age scope, sample size, follow-up, and comments.

| Item | Title | Journal | Country | Type | Scale Used | Citation | Year | Dimensions | | | | FL Definition by Author | Statistic Test Validity/ Reliability | Contribution Statement | Age Scope | Sample Size | Follow-Up Time, If Applicable | Comments |
|---|---|---|---|---|---|---|---|---|---|---|---|---|---|---|---|---|---|---|
| | | | | | | | | FK | FB | FA | # | | | | | | | |
| 1 | Bibliometric mapping of research trends on financial behavior for sustainability | *Sustainability* | Various countries | Review | N/A | [86] | 2022 | X | X | X | 3 | FL is associated with financial education socialization processes, which include financial learning, attitudes, and behavior and contribute to sustainable development. | None | This article presents a global empirical view of financial behavior studies associated with the Sustainable Development Goals (SDGs) in the period between 1992 and 2020. | N/A | N/A | N/A | Financial education constitutes a differentiating and necessary element to maintain personal finances; however, it has not achieved high prestige in the international scientific community. |
| 2 | Financial literacy and the use of credit cards in Mexico | *Journal of International Studies* | Mexico | Scale | Financial knowledge. | [87] | 2021 | X | X | - | 2 | FL is studied as knowledge and behaviors that enable one to make financial decisions in various contexts, such as the proper use of credit. | None | This research "identifies the level of financial literacy of Mexican cardholders and its relationship with sociodemographic variables." | 15–74 | 2170 | N/A | It was found that Mexican cardholders have a higher level of FL than the general population and women have lower FL than men. |
| 3 | The impact of attitudes on behavioral change: A multilevel analysis of predictors of changes in consumer behavior | *Revista Latinoamericana de Psicología* | Chile | Intervention | Attitudes Toward Purchasing Scale. Values-oriented Materialism Scale. Adult's Economic Literacy Scale. Self-Discrepancy Scale. Consumption Habits and Behaviors Scale. | [27] | 2021 | X | X | X | 3 | FL is approached with a tutoring methodology that aims to provide specific finance knowledge and develop practical skills and positive attitudes toward the use of resources. | Test–retest reliability $W > 0.677$ | This paper identifies the predictor variables related to responsible consumption. | Students | 110 | N/A | Rationality and centrality are significant predictors of behavioral changes in purchasing. |

**Table 1.** *Cont.*

| Item | Title | Journal | Country | Type | Scale Used | Citation | Year | Dimensions | | | | FL Definition by Author | Statistic Test Validity/ Reliability | Contribution Statement | Age Scope | Sample Size | Follow-Up Time, If Applicable | Comments |
|---|---|---|---|---|---|---|---|---|---|---|---|---|---|---|---|---|---|---|
| | | | | | | | | FK | FB | FA | # | | | | | | | |
| 4 | Multiagent Intelligent Tutoring System for Financial Literacy | *International Journal of Web-Based Learning and Teaching Technologies* | Brazil | Intervention | N/A | [28] | 2021 | X | X | - | 2 | FL is a process that is made up of several areas of knowledge, which can range from basic to advanced. | None | The authors propose a prototype that can integrate artificial intelligence tools with the aim of specifying a pattern in middle and high school or "ensino medio" and the learning of new knowledge. | University students | 100 | 4 months | This article represents innovative multidisciplinary research. |
| 5 | Healthily Crazy Business! Solidarity Economy and Financial Education as Emancipation Tools for the Mentally | *Innovar* | Brazil | Intervention | N/A | [29] | 2021 | X | X | - | 2 | FL is defined by the authors as the ability to manage money based on the financial knowledge they possess. | None | The authors provide a thoughtful critique of a program focused on the solidarity economy that can give people suffering from mental illness an opportunity to integrate into society. | Adults | N/A | 3 months | The authors implement a program designed so that people with mental illnesses can have control over their spending. |

Table 1. *Cont.*

| Item | Title | Journal | Country | Type | Scale Used | Citation | Year | Dimensions | | | | FL Definition by Author | Statistic Test Validity/ Reliability | Contribution Statement | Age Scope | Sample Size | Follow-Up Time, If Applicable | Comments |
|------|-------|---------|---------|------|-----------|----------|------|----|----|----|---|------------------------|--------------------------------------|------------------------|-----------|-------------|------------------------------|----------|
| | | | | | | | | FK | FB | FA | # | | | | | | | |
| 6 | Choice architecture improves pension selection | *Applied Economics* | Chile | Intervention | N/A | [30] | 2021 | X | - | X | 2 | Financial literacy is considered basic financial knowledge related to compound interest rates and risk management. | None | The authors proposed a supply design for pension providers that may generate higher levels of financial wellbeing for individuals in Chile. | Females and males between the ages of 55 and 70 | 1041 | N/A | The positive effects on respondents of reducing information demands (eliminating the risk information from the comparisons) and adopting a loss frame were particularly evident for respondents with low levels of financial literacy. |
| 7 | Exploratory experience of validation of an instrument on the level of financial literacy in the millennial generation | *Revista de Métodos Cuantitativos para la Economia y la Empresa* | Various countries | Scale | Test de Alfabetización Económica para Adultos. | [73] | 2021 | X | X | - | 2 | Financial literacy includes knowledge related to budget management, money control, period planning, and the choice of financial products that provide the most significant benefit. | Cronbach's alphas = 0.681 | The authors detail the exploratory experience regarding validating an instrument to measure the level of financial culture in millennials. | Millennials aged 18 to 30 | 206 | N/A | The content of the instrument can be used for data collection. |
| 8 | Financial Perceptions and Skills Among University Students | *Formacion Universitaria* | Colombia | Scale | Elaborated by authors. | [23] | 2021 | X | - | - | 1 | Financial literacy is understood as knowledge associated with finance. | Cronbach's alphas = 0.89 | The authors explain that students may respond favorably to financial problems but have weaknesses regarding financial skills. | University students | 307 | N/A | It is possible to identify that most students give significant importance (95.8%) to topics such as money, savings, investment, rate of return, etc. (68.8% strongly agree and 27.0% agree). |

**Table 1.** *Cont.*

| Item | Title | Journal | Country | Type | Scale Used | Citation | Year | Dimensions | | | | FL Definition by Author | Statistic Test Validity/ Reliability | Contribution Statement | Age Scope | Sample Size | Follow-Up Time, If Applicable | Comments |
|---|---|---|---|---|---|---|---|---|---|---|---|---|---|---|---|---|---|---|
| | | | | | | | | FK | FB | FA | # | | | | | | | |
| 9 | The financial wellbeing of the beneficiaries of the minha casa minha vida program: Perception and antecedents | *Revista de Administracao Mackenzie* | Brazil | Scale | Consumer Financial Protection Bureau. Financial wellbeing scale. | [88] | 2021 | X | X | X | 3 | The authors consider financial literacy a combination of knowledge, attitudes, and behaviors necessary to make correct financial decisions. | None | The authors replicate the methodology proposed by the Consumer Financial Protection Bureau (CFPB) in Brazil to demonstrate its effectiveness in terms of financial wellbeing. | Beneficiaries of the three funding ranges of the "Minha Casa Minha Vida" program | 561 | N/A | The BEF measure proposed by the Consumer Financial Protection Bureau seems adequate in the Brazilian context. |
| 10 | Improving the level of financial literacy and the influence of the cognitive ability in this process | *Journal of Behavioral and Experimental Economics* | Brazil | Intervention | N/A | [34] | 2021 | X | X | X | 3 | The authors consider financial literacy a combination of financial knowledge, financial attitude, and financial behavior. | None | The authors conclude that there is no consensus about the effect that personal finance courses have on financial literacy. | University students | 517 | N/A | It includes two studies in one paper, both using different scales. The sample size corresponds to the first model. |
| 11 | Inclusive health and life insurance adoption: An empirical study in Guatemala | *Review of Development Economics* | Guatemala | Scale | The Short Grit Scale. The Big Five personality traits. Scale for neuroticism. Risk preferences. Numerical abilities. | [49] | 2021 | X | X | - | 2 | Financial literacy has conceived the knowledge of basic economic and financial concepts. | None | The authors analyze health insurance decisions in a context motivated by combining new personality theories with health theories. | Above 18 | 701 | N/A | N/A |

**Table 1.** *Cont.*

| Item | Title | Journal | Country | Type | Scale Used | Citation | Year | Dimensions | | | | FL Definition by Author | Statistic Test Validity/ Reliability | Contribution Statement | Age Scope | Sample Size | Follow-Up Time, If Applicable | Comments |
|------|-------|---------|---------|------|-----------|----------|------|----|----|----|---|------------------------|---------------------------------|------------------------|-----------|-------------|------------------------------|----------|
| | | | | | | | | **FK** | **FB** | **FA** | **#** | | | | | | | |
| 12 | On the financial literacy, indebtedness, and wealth of Colombian households | *Review of Development Economics* | Colombia | Longitudinal | Adapted from: Comprehensive Household Survey and the Survey of Financial Burden and Literacy of Bogotá. | [80] | 2021 | X | X | - | 2 | The authors review the definitions of financial literacy considering five categories: knowledge of financial concepts, ability to communicate them, aptitude in the management of personal finances, skills in making financial decisions, and confidence in planning for future needs. | None | Analysis of econometric models proposed by the authors, where financial literacy estimates depend on debt and wealth indicators. | Above 18 | 83,100 | 2011–2015 | |
| 13 | Choosing the highest annuity payout: the role of inter-mediation and firm reputation | *Geneva Papers on Risk and Insurance: Issues and Practice* | Chile | Longitudinal | N/A | [82] | 2021 | X | - | - | 1 | Financial literacy is considered the ability to make better financial decisions. | None | The authors found a positive correlation between the choice of annuity payment and the advice of independent sales agents, taking a sample from 2008 to 2018. | Retirees | 199,114 | 10 years | N/A |

**Table 1.** *Cont.*

| Item | Title | Journal | Country | Type | Scale Used | Citation | Year | Dimensions | | | | FL Definition by Author | Statistic Test Validity/ Reliability | Contribution Statement | Age Scope | Sample Size | Follow-Up Time, If Applicable | Comments |
|------|-------|---------|---------|------|-----------|----------|------|----|----|----|---|-------------------------|--------------------------------------|------------------------|-----------|-------------|-------------------------------|----------|
| | | | | | | | | FK | FB | FA | # | | | | | | | |
| 14 | Emergent curriculum in basic education for the new normality in Peru: orientations proposed from mathematics education | *Educational Studies in Mathematics* | Peru | Review | N/A | [89] | 2021 | X | - | - | 1 | Financial literacy is defined as the understanding of financial concepts and risks to improve the financial wellbeing of individuals and society. | None | The author proposes a new mathematics curriculum that includes ethical, political, financial, and statistical issues and a problem-solving approach to facing the new normal in education. | N/A | N/A | N/A | N/A |
| 15 | Financial Literacy and the Perceived Value of Stress Testing: An Experiment Using Students in Brazil | *Emerging Markets Finance and Trade* | Brazil | Intervention | N/A | [74] | 2021 | X | X | X | 3 | Financial literacy is measured as the knowledge of interest rate accrual, inflation and purchasing power, and risk diversification. | None | This study contributes to the literature by providing empirical evidence from the consumer's point of view, on the value of communicating routine tests of bank stress. | Students | 375 | N/A | The authors found that "respondents with higher financial literacy levels are more conservative and replace more profitable banks with less profitable banks that perform and communicate stress testing routines to the public." |

**Table 1.** *Cont.*

| Item | Title | Journal | Country | Type | Scale Used | Citation | Year | Dimensions | | | | FL Definition by Author | Statistic Test Validity/ Reliability | Contribution Statement | Age Scope | Sample Size | Follow-Up Time, If Applicable | Comments |
|---|---|---|---|---|---|---|---|---|---|---|---|---|---|---|---|---|---|---|
| | | | | | | | | FK | FB | FA | # | | | | | | | |
| 16 | Parents Influence Responsible Credit Use in Young Adults: Empirical Evidence from the United States, France, and Brazil | *Journal of Family and Economic Issues* | Various countries | Scale | Elaborated by authors. | [90] | 2021 | X | X | X | 3 | Financial literacy is considered the conceptual result of financial education. | Cronbach's alpha > 0.69; AVE > 0.43 | The model validated during the research examines the effect of using credit cards on financial wellbeing, which can also be affected by social comparison and financial self-confidence. | Young adults | 1458 | N/A | The authors found evidence that men are more dependent on parental education than women. |
| 17 | Technological gender gap in university financial education in Mexico | *Revista Venezolana de Gerencia* | Mexico | Scale | Elaborated by authors. | [24] | 2021 | X | X | - | 2 | The authors define financial literacy as the set of knowledge and behaviors that produce effective financial education. | Cronbach's alpha =0.7 | This study provides an analysis of the technological gap between the levels of financial education of university students. | University students | 3215 | N/A | The authors conclude that young people have more significant financial knowledge in technological matters due to the close relationship to technology development. |
| 18 | Exploring the antecedents of retail banks' reputation in low-bankarization markets: brand equity, value co-creation and brand experience | *International Journal of Bank Marketing* | Ecuador | Scale | Elaborated by authors. | [91] | 2021 | X | - | - | 1 | Financial literacy is determined by knowledge about financial topics. | None | The author provides an original perspective that helps banks improve their reputation through strategies that focus on customer–business interaction and branding. | N/A | N/A | N/A | N/A |

Table 1. *Cont.*

| Item | Title | Journal | Country | Type | Scale Used | Citation | Year | Dimensions | | | | FL Definition by Author | Statistic Test Validity/ Reliability | Contribution Statement | Age Scope | Sample Size | Follow-Up Time, If Applicable | Comments |
|---|---|---|---|---|---|---|---|---|---|---|---|---|---|---|---|---|---|---|
| | | | | | | | | FK | FB | FA | # | | | | | | | |
| 19 | Financial Citizenship Perception (FCP) Scale: proposition and validation of a measure | *International Journal of Bank Marketing* | Brazil | Scale | Elaborated by authors. Financial citizenship perception scale. | [70] | 2021 | X | - | - | 1 | The authors consider financial literacy as the knowledge of financial concepts and products. | None | The authors propose a scale that measures the perception of financial citizenship concerning three dimensions: financial inclusion, financial protection, and financial literacy. | N/A | N/A | N/A | The scale proposed in the study allows for developing an indicator that defines whether an individual has a high or low level of financial citizenship, represented by the dimensions of financial inclusion, financial protection, and financial literacy. |
| 20 | The effect of financial literacy and gender on retirement planning among young adults | *International Journal of Bank Marketing* | Mexico | Longitudinal | N/A | [31] | 2021 | X | X | X | 3 | Financial literacy is comprised of individuals' financial inclusion, attitudes, knowledge, behavior, occupation, and family traits. | None | This study provides long-term information on the investment plans of young Mexicans, focusing on financial education and gender. | Young adults | N/A | N/A | The authors confirm that the most financially knowledgeable individuals are less likely to pursue passive strategies, while financial behavior and inclusion are associated with active planning. |
| 21 | Economic literacy in the most vulnerable population of Barranquilla | *Revista Venezolana de Gerencia* | Colombia | Scale | Elaborated by authors. | [92] | 2021 | X | X | X | 3 | The authors take the definition of financial literacy as "the ability to use knowledge and skills to manage financial resources effectively". | N/A | The authors analyze the economic literacy in terms of consumption and debt in vulnerable families in Barranquilla, Colombia | Families | 18 | N/A | The consumption rate of the people in the sample sometimes exceeds their income since they would like to have a better standard of living; however, the high cost of indebtedness and the dissuasive risk rating systems limit their access to credit. |

**Table 1.** *Cont.*

| Item | Title | Journal | Country | Type | Scale Used | Citation | Year | Dimensions | | | | FL Definition by Author | Statistic Test Validity/ Reliability | Contribution Statement | Age Scope | Sample Size | Follow-Up Time, If Applicable | Comments |
|------|-------|---------|---------|------|------------|----------|------|----|----|----|---|--------------------------|--------------------------------------|------------------------|-----------|-------------|-------------------------------|----------|
| | | | | | | | | FK | FB | FA | # | | | | | | | |
| 22 | Financial attitude, financial behavior, and financial knowledge, in Mexico | *Cuadernos de Economía* | Mexico | Scale | Financial attitude. Financial behavior. Financial knowledge. | [22] | 2021 | X | X | X | 3 | Financial literacy is described as knowledge and skills that make it easier to manage financial resources. | N/A | The authors estimate the levels of financial attitude, behavior, and knowledge in Mexico and investigate their relationship with sociodemographic variables to contribute to the design and implementation of better-targeted private initiatives and government programs. | 18–70 | 12,446 | N/A | The results confirm that financial literacy is low, is mainly affected by education, and that a gender gap exists. |
| 23 | Financial Literacy among millennials in Ciudad Victoria, Tamaulipas, Mexico | *Estudios Gerenciales* | Mexico | Scale | Financial knowledge. Elaborated by authors, numerical ability. | [42] | 2021 | X | X | X | 3 | The author defines financial literacy as "knowledge and skills that enable people to manage money more efficiently, make better financial decisions, and make planning for their future easier." | N/A | This work evaluates measurement indexes to determine the most convenient for studying financial literacy among millennials from Ciudad Victoria, Tamaulipas, Mexico, to contribute to designing financial education strategies that promote investment and entrepreneurship among young people. | 15–29 | 1529 | N/A | The sample selection procedure constitutes the main limitation of this research, given that the results are difficult to generalize. |

**Table 1.** *Cont.*

| Item | Title | Journal | Country | Type | Scale Used | Citation | Year | Dimensions | | | | FL Definition by Author | Statistic Test Validity/ Reliability | Contribution Statement | Age Scope | Sample Size | Follow-Up Time, If Applicable | Comments |
|---|---|---|---|---|---|---|---|---|---|---|---|---|---|---|---|---|---|---|
| | | | | | | | | **FK** | **FB** | **FA** | **#** | | | | | | | |
| 24 | Multinomial logistic regression to estimate the financial education and financial knowledge of university students in Chile | *Information* | Chile | Scale | Survey of Measurement of Financial Capabilities | [93] | 2021 | X | X | X | 3 | The authors mention that "financial literacy is characterized by an initial stage that seeks to know and relate the essential financial concepts for personal finance". | N/A | This study analyzes university students' knowledge, behavior, and attitudes toward financial education by fitting a multinomial logistic regression model. | University students | 410 | N/A | The fact that this study is aimed at young people is relevant since they will be the decision makers in the coming decades and should be as prepared as possible. |
| 25 | Effects of Financial Crises on Latin American Economies-Colombia Case and the SMEs | *IBIMA Business Review* | Colombia | Review | N/A | [51] | 2020 | X | - | - | 1 | Financial literacy is approached as the ability of SMEs when defining strategies to obtain good economic results and avoid unfavorable outcomes. | N/A | This article reviews what happened in the financial crises of the Colombian economy, identifying the effects on small and medium-sized companies and the potential of financial literacy as a tool to cushion the potent effects of these crises. | N/A | N/A | N/A | In addition to affecting the Colombian economy, the crises negatively impacted the most vulnerable companies, which are the smallest. |

**Table 1.** *Cont.*

| Item | Title | Journal | Country | Type | Scale Used | Citation | Year | Dimensions | | | | FL Definition by Author | Statistic Test Validity/ Reliability | Contribution Statement | Age Scope | Sample Size | Follow-Up Time, If Applicable | Comments |
|------|-------|---------|---------|------|------------|----------|------|----|----|----|---|-------------------------|--------------------------------------|-----------------------|-----------|-------------|------------------------------|----------|
| | | | | | | | | FK | FB | FA | # | | | | | | | |
| 26 | Choice of pension management fees and effects on pension wealth | *Journal of Economic Behavior and Organization* | Peru | Intervention | N/A | [53] | 2020 | X | - | - | 1 | The authors measure financial literacy as how actively individuals manage their pension fund portfolios. | None | The authors show the adverse effect of Peru's pension fund quota reform, which came into effect in 2013 on pension wealth. | Non-retired population from SBS's administrative registers as of December 2016 | 64,588 | N/A | The results suggest that financial literacy plays an essential role in making sound decisions about pension fund fees. Unfortunately, Peru has deficient financial literacy levels, and this limitation was not considered in the policy design. |
| 27 | Does formal and business education expand the levels of financial education? | *International Journal of Social Economics* | Brazil | Intervention | Financial literacy thermometer. Financial citizenship series. Personal finance index. | [32] | 2020 | X | X | X | 3 | Financial literacy is measured with a financial literacy thermometer that considers the three dimensions proposed by the OECD. | None | The study shows that knowledge formally acquired by individuals can raise levels of financial literacy. | Under-graduate students | 285 | N/A | The target population comprises undergraduate students in Business Administration from the Federal University of Santa Catarina. |
| 28 | Financial illiteracy and customer credit history | *Revista Brasileira de Gestao de Negocios* | Brazil | Intervention | National survey on financial inclusion in Brazil and information on financial instrument usage and spending. | [52] | 2020 | X | - | - | 1 | The author used capitalization bonds as a proxy for financial literacy. | None | A proxy for illiteracy correlated with the investment and debt decisions of families is proposed. | Brazilian house-holds | 2885 | N/A | They use the National Survey on Financial Inclusion in Brazil and information on financial instrument usage and spending provided by the head of the household. |

**Table 1.** *Cont.*

| Item | Title | Journal | Country | Type | Scale Used | Citation | Year | Dimensions | | | | FL Definition by Author | Statistic Test Validity/ Reliability | Contribution Statement | Age Scope | Sample Size | Follow-Up Time, If Applicable | Comments |
|---|---|---|---|---|---|---|---|---|---|---|---|---|---|---|---|---|---|---|
| | | | | | | | | FK | FB | FA | # | | | | | | | |
| 29 | Financial literacy level on college students: A comparative descriptive analysis between Mexico and Colombia | *European Journal of Contemporary Education* | Various countries | Scale | Retirement planning scale. Inflation scale. Numeracy scale. Insurance scale. Huston scale about pop credit. Volpe scale about saving and investment. Scale about risk diversification. | [94] | 2020 | X | X | X | 3 | The measure of the knowledge level about the financial information needed for a person to make trustworthy financial choices. | None | This paper proposes a scale that considers seven financial issues that measure financial literacy, and it is possible to replicate it in other populations. | College students | 224 | N/A | We must point out that college students have an ideal opportunity to focus their efforts on financial literacy since they are at a crucial stage of their lives when they are just entering the workforce or will do so soon. |
| 30 | Is the dosed video-vignettes intervention more effective with a longer-lasting effect? A financial literacy Study. | *ACM International Conference Proceeding Series* | Ecuador | Longitudinal | FL scale. | [48] | 2020 | X | X | X | 3 | Financial literacy measures the financial knowledge, behavior, and attitudes that improve key financial decisions. | None | This article identifies the effect of video vignettes as an intervention, which can have a lasting effect and increase financial literacy. | University students | 111 | 6 months | The response time was slightly longer than the source material publication from Méndez et al., 2019; 25 min for the pretest, 15 min for the post-test, and 10 min on average for every video. Once again, the proximity of these events makes the students answer faster because they recognize the survey they filled out days ago. |

**Table 1.** *Cont.*

| Item | Title | Journal | Country | Type | Scale Used | Citation | Year | Dimensions | | | | FL Definition by Author | Statistic Test Validity/ Reliability | Contribution Statement | Age Scope | Sample Size | Follow-Up Time, If Applicable | Comments |
|---|---|---|---|---|---|---|---|---|---|---|---|---|---|---|---|---|---|---|
| | | | | | | | | FK | FB | FA | # | | | | | | | |
| 31 | Knowledge and application toward financial topics in high school students: A parametric study | *European Journal of Educational Research* | Mexico | Scale | Adapted from Financial Education Scale. | [39] | 2020 | X | - | - | 1 | A measure of a person's competence to comprehend and apply financial information. | Cronbach's alphas = 0.860 with 34 items | The study shows how certain factors influence financial knowledge and financial techniques when handling different financial products. | High school students | 305 | N/A | 333 high school students were surveyed face to face, and only 305 were validated. The internal consistency of Cronbach's alpha of the scale was $\alpha = 0.860$ (34 items) and $\alpha = 0.855$ (7 dimensions). |
| 32 | The role of cognitive abilities on financial literacy: New experimental evidence | *Journal of Behavioral and Experimental Economics* | Colombia | Intervention | Adapted from survey to measure financial literacy. | [46] | 2020 | X | X | X | 3 | It is the set of abilities and knowledge applied in the decision-making process that leads to a sound individual financial status. | None | This article contributes to the contrast between the recent empirical literature and surveys to show the relationship between cognitive ability and financial literacy. | Under-graduate students | 195 | N/A | The results show that people with more significant cognitive capacity are more likely to have better financial literacy. This may be due to various reasons such as risk aversion, biases, etc. |

**Table 1.** *Cont.*

| Item | Title | Journal | Country | Type | Scale Used | Citation | Year | Dimensions | | | | FL Definition by Author | Statistic Test Validity/ Reliability | Contribution Statement | Age Scope | Sample Size | Follow-Up Time, If Applicable | Comments |
|---|---|---|---|---|---|---|---|---|---|---|---|---|---|---|---|---|---|---|
| | | | | | | | | FK | FB | FA | # | | | | | | | |
| 33 | Adaptation and validation of the economic and financial literacy test for Chilean secondary students | *Revista Latinoamericana de Psicologia* | Chile | Scale | Adapted from financial literacy test (TAEF-E). Scale of susceptibility to the influence of peers in consumption. Scale of youth materialism. | [95] | 2019 | X | X | - | 2 | It is explained as the capacity to systematically understand the social and economic model to include and use basic economic definitions and interpret situations and economic policies that may be implicit. | GFI, CFI, NFI, and TLI >= 0.95 RMSEA < 0.08 | The authors propose an instrument to measure the financial and economic literacy of high school students and evaluate the scale's psychometric properties by checking its reliability. | High school students | 811 | N/A | The authors used two-stage sampling: the first was non-probabilistic and purposive, and a list of urban, co-ed high schools was compiled. They used a probability sample with a 5% error rate and a 95% confidence level in the second stage. |
| 34 | Determinants of knowledge of personal loans' total costs: How to price con-sciousness, financial literacy, purchase recency, and frequency work together | *Journal of Business Research* | Chile | Scale | Scale of purchase frequency, brand costumer. Scale of purchase recency. Scale of customer satisfaction. Scale of price consciousness. Scale about price–quality cue, price advertising exposure. Adapted from Kau scale, brand credibility. | [37] | 2019 | X | - | - | 1 | Financial literacy is measured as the ability to perform simple calculations, understand how compound interest works, and the effect of inflation. | Cronbach's alphas > 0.7 | The authors conclude that the information on loans is very complex, and this implies that frequent and more knowledgeable customers do not necessarily know the total costs of the loans. | Those who plan to acquire a personal loan within the next three months and current cus-tomers of personal loans that are aged 18 years and over | 392 | N/A | Future research could employ other scales to include different sources of quality of a personal loan. |

**Table 1.** *Cont.*

| Item | Title | Journal | Country | Type | Scale Used | Citation | Year | Dimensions | | | | FL Definition by Author | Statistic Test Validity/ Reliability | Contribution Statement | Age Scope | Sample Size | Follow-Up Time, If Applicable | Comments |
|---|---|---|---|---|---|---|---|---|---|---|---|---|---|---|---|---|---|---|
| | | | | | | | | FK | FB | FA | # | | | | | | | |
| 35 | Financial Literacy in Brazil—do knowledge and self-confidence relate to behavior? | *RASP Management Journal* | Brazil | Scale | Adapted from International Survey of Adult Financial Literacy Competencies. | [36] | 2019 | X | X | X | 3 | Financial literacy is presented as a model where actual financial knowledge is a predictor of perceived financial knowledge and financial attitude, and all three argue that financial behavior is the most critical variable of interest for this analysis. | None | This article makes a methodological contribution channeled in the knowledge of multiple individuals with different levels of financial literacy. It also has a theoretical gift with which it highlights the relationship between self-confidence and the behavior of individuals. | N/A | 1487 | N/A | Despite these results, the total sample was heterogeneous regarding people's actual financial knowledge and self-confidence levels, justifying multi-group analyses to investigate any differences in the study's conceptual model. |
| 36 | Financial Literacy of "telebachillerato" students: A study of perception, usefulness, and application of financial tools | *International Journal of Education and Practice* | Mexico | Scale | Adapted from Financial Education Test. | [40] | 2019 | X | - | - | 1 | Financial literacy analyzes the knowledge of individuals regarding the management of their income. | Cronbach's alphas > 0.8 | This study analyzes Mexican students' perception of financial tools to determine what motivates them to continue studying. | Students | 368 | N/A | N/A |
| 37 | The impact of video vignettes to enhance the financial literacy level of Ecuadorian university students | *ACM International Conference Proceeding Series* | Ecuador | Intervention | Elaborated by authors. | [47] | 2019 | X | X | X | 3 | Financial literacy is a measure of the financial knowledge, behavior, and attitudes that improve critical financial decision making. | None | This article proposes specialized material as an interactive tool that improves the level of financial education of university students in Ecuador. | University students | 100 | N/A | This study analyzes the three dimensions of the financial literature: knowledge, behavior, and attitude. |

**Table 1.** *Cont.*

| Item | Title | Journal | Country | Type | Scale Used | Citation | Year | Dimensions | | | | FL Definition by Author | Statistic Test Validity/ Reliability | Contribution Statement | Age Scope | Sample Size | Follow-Up Time, If Applicable | Comments |
|---|---|---|---|---|---|---|---|---|---|---|---|---|---|---|---|---|---|---|
| | | | | | | | | FK | FB | FA | # | | | | | | | |
| 38 | Determinants of financial deepening in Mexico: A dynamic panel data approach | *Revista de Métodos Cuantitativos para la Economía y la Empresa* | Mexico | Longitudinal | N/A | [83] | 2019 | X | - | - | 1 | The financial literacy wing includes the study of financial productivity at the macroeconomic level. | None | The authors propose a panel data model that had not been used before to address Mexico's financial determinants such as the rule of law, banking regulation, propensity to save, etc. | Federal Entities | 32 | 5 years | This document shows that the main determinants of financial deepening are the rule of law or institutions, bank regulation, banking competition, formal labor, saving propensity, and financial education. |
| 39 | The antecedents and consequences of financial literacy: a meta-analysis | *International Journal of Bank Marketing* | Brazil | Review | N/A | [33] | 2019 | X | X | X | 3 | The competence of using knowledge and skills to administer financial means effectively for a lifetime of financial wellness. | None | This research demonstrates the impact of the background and consequences of financial education through a meta-analysis, offering, in turn, empirical generalizations about the effects investigated. | N/A | 690 | N/A | The consequences of financial literacy were the behavior of incurring avoidable credit and checking fees, credit score, and the willingness to take investment risks. |

**Table 1.** *Cont.*

| Item | Title | Journal | Country | Type | Scale Used | Citation | Year | Dimensions | | | | FL Definition by Author | Statistic Test Validity/ Reliability | Contribution Statement | Age Scope | Sample Size | Follow-Up Time, If Applicable | Comments |
|---|---|---|---|---|---|---|---|---|---|---|---|---|---|---|---|---|---|---|
| | | | | | | | | FK | FB | FA | # | | | | | | | |
| 40 | Financial Literacy and Money Script: A Caribbean Perspective | *Springer Nature* | Trinidad and To-bago | Review | N/A | [77] | 2018 | X | X | - | 2 | Financial literacy is considered essential for decision making, given the level of knowledge. | None | This paper shows financial literacy components by addressing debt concerns, overreach in credit card use, risk management, and retirement. | N/A | N/A | N/A | This paper covers a wide range of topics. It assures the reader that understanding one's money script and making changes (if necessary) will result in more effective and responsible management and handling of one's financial affairs. |
| 41 | A financial literacy model for university students | *Individual Behaviors and Technologies for Financial Innovations* | Brazil | Scale | Factor from the average response of two groups of multiple-choice questions to evaluate the academic level of financial knowledge. | [2] | 2018 | X | X | X | 3 | The dominance of a set of knowledge, attitudes, and behaviors has taken on an essential role in enabling people to make assertive decisions as they try to obtain financial wellness. | RMSEA > 0.08 GFI, CFI, NFI, and TLI < 0.95 | The authors demonstrate the effectiveness of a metric that encompasses the dimensions of financial literacy: financial knowledge, behavior, and attitude. | Students | 534 | N/A | The data were collected only in Brazil, presenting explicit peculiarities, such as an economic structure for services. Therefore, different countries should be researched using a larger sample. |

**Table 1.** *Cont.*

| Item | Title | Journal | Country | Type | Scale Used | Citation | Year | Dimensions | | | | FL Definition by Author | Statistic Test Validity/ Reliability | Contribution Statement | Age Scope | Sample Size | Follow-Up Time, If Applicable | Comments |
|------|-------|---------|---------|------|-----------|----------|------|----|----|----|---|--------------------------|--------------------------------------|------------------------|-----------|-------------|-------------------------------|----------|
| | | | | | | | | FK | FB | FA | # | | | | | | | |
| 42 | Demystifying Financial Literacy: a behavioral perspective analysis | *Management Research Review* | Brazil | Scale | Scale to measure financial knowledge (multiple choice). Scale of propensity to indebtedness. | [96] | 2018 | X | X | X | 3 | Financial literacy has been recognized as a key competency that affects behavioral factors: materialism, compulsive shopping, and a propensity to borrow. | Cronbach's alphas > 0.7 RMSEA < 0.08 | This paper measures the impact of financial literacy on behavioral factors; that is, the relationship between these components while not studying them individually as is usually done. | 35 | 2487 | N/A | The main findings showed that financial literacy's impact on compulsive buying behavior was the greatest of the direct relationships proposed and the total effects of financial literacy on behavioral aspects. |
| 43 | Financial Literacy and informal loan | *Individual Behaviors and Technologies for Financial Innovations* | Brazil | Intervention | National survey on financial inclusion and the use of banking correspondents in Brazil adapted from Bankable Frontier Associates scale. | [54] | 2018 | X | X | X | 3 | The union of awareness, knowledge, skill, attitude, and behavior is necessary to make smart financial choices and ultimately achieve individual financial wellness. | None | This study focuses on analyzing the impact of financial education on informal loans that friends, acquaintances, or non-formal lenders can give. | Families | 2023 | N/A | The proxy adapted for financial literacy is the consumption of a particular financial product called a capitalization bond. |

**Table 1.** *Cont.*

| Item | Title | Journal | Country | Type | Scale Used | Citation | Year | Dimensions | | | | FL Definition by Author | Statistic Test Validity/ Reliability | Contribution Statement | Age Scope | Sample Size | Follow-Up Time, If Applicable | Comments |
|------|-------|---------|---------|------|-----------|----------|------|----|----|----|---|------------------------|--------------------------------------|------------------------|-----------|-------------|-------------------------------|----------|
| | | | | | | | | FK | FB | FA | # | | | | | | | |
| 44 | How well do women do when it comes to financial literacy? The proposition of an indicator and analysis of gender differences | *Journal of Behavioral and Experimental Finance* | Brazil | Scale | Scale of financial literacy level. Patrick scale, financial attitude, financial behavior, financial knowledge. | [17] | 2018 | X | X | X | 3 | The unification of the essential awareness, knowledge, skill, attitude, and behavior needed to make healthy decisions that can lead to the achievement of individual financial wellness. | Cronbach's alphas = 0.6 RMSEA = 0.08 | The authors develop an indicator that measures the financial literacy level of adults and study the knowledge gap between men and women. | Above 18 | 2485 | N/A | The main results show that most individuals have a low level of financial literacy across both genders. However, the proportion of men is higher among those with a high level of financial literacy. |
| 45 | Micro-entrepreneurship Debt Level and Access to Credit: Short-Term Impacts of a Financial Literacy Program | *European Journal of Development Research* | Chile | Intervention | N/A | [43] | 2018 | X | - | - | 1 | The financial literacy of microentrepreneurs influences their decisions about loans and investments. | None | First, the authors identify the impact of the program applied to households on the formal credit system. Second, the program is carried out on a large scale, guaranteeing external validity, which is more accurate than other small programs. | Participants from households and the micro-entrepreneurship component | 4526 | N/A | The authors concluded that the program tends to decrease debt levels in the short run while increasing the probability of having formal debt. In addition, the program had effects in parts of the country where the take-up was higher, and the implementation was smoother. |

**Table 1.** *Cont.*

| Item | Title | Journal | Country | Type | Scale Used | Citation | Year | Dimensions | | | | FL Definition by Author | Statistic Test Validity/ Reliability | Contribution Statement | Age Scope | Sample Size | Follow-Up Time, If Applicable | Comments |
|---|---|---|---|---|---|---|---|---|---|---|---|---|---|---|---|---|---|---|
| | | | | | | | | FK | FB | FA | # | | | | | | | |
| 46 | How numeracy mediates cash flow format preferences: A worldwide study | *International Journal of Management Education* | Chile | Intervention | N/A | [75] | 2018 | - | X | X | 2 | For financial literacy, numeracy plays a key role in student behavior and decisions. | None | Contribution focused on direct and indirect effects on cash flow statements to make loan decisions. | Students | 688 | N/A | Future research could shed more light on this discovery, which has essential policy ramifications for banks, accounting firms, and accounting functions in larger firms. |
| 47 | Financing millennials in developing economies: Banking strategies for undergraduate students | *Marketing Techniques for Financial Inclusion and Development* | Mexico | Review | N/A | [78] | 2018 | X | X | X | 3 | Financial literacy encompasses financial knowledge, attitudes, behaviors, and inclusion. | None | Analysis of the behavior and attitudes of millennials regarding their access to savings and bank credit in contrast to traditional sources of financing and their perspective on the issue of financial inclusion. | Millennials aged 18 to 24 | 50 | N/A | N/A |
| 48 | Lower financial literacy induces the use of informal loans | *RAE Revista de Administracao de Empresas* | Brazil | Intervention | N/A | [21] | 2018 | X | X | - | 2 | Financial literacy in households is directly related to the propensity to make loans. | None | This study focuses on formal lending markets and looks at the impact of financial literacy on informal lending. | Families | 2023 | N/A | The results suggest that financial literacy's relevance to informal loans may exceed that of formal credit channels. |

**Table 1.** *Cont.*

| Item | Title | Journal | Country | Type | Scale Used | Citation | Year | Dimensions | | | | FL Definition by Author | Statistic Test Validity/ Reliability | Contribution Statement | Age Scope | Sample Size | Follow-Up Time, If Applicable | Comments |
|---|---|---|---|---|---|---|---|---|---|---|---|---|---|---|---|---|---|---|
| | | | | | | | | FK | FB | FA | # | | | | | | | |
| 49 | Profiles of saving and payment of the debt in the life cycle of Mexican households | *Trimestre Economico* | Mexico | Scale | Encuesta nacional de ingresos y gastos de los hogares. | [72] | 2018 | - | X | - | 1 | Financial literacy in households determines behavior and their saving, credit, and investment decisions. | None | This paper provides a semi-parameter estimate of individuals' savings and debt patterns throughout their life cycle. | Families | 29,468 | 2000–2014 | The ENIGH captures the evolution of the primary income and expenditure indicators of households in Mexico; it also collects information about the characteristics of the dwellings, their members, the household equipment, the occupation status of the individuals, their level of education, etc. The samples ranged from 9002 to 29,468, but we consider the largest. |
| 50 | The context of financial innovations. | *Individual Behaviors and Technologies for Financial Innovations* | Brazil | Review | N/A | [79] | 2018 | - | X | - | 1 | Financial literacy in companies is influenced by innovation and, in turn, affects consumer behavior. | None | This paper provides an in-depth and detailed analysis of the importance of individual behavior and financial innovation. | N/A | N/A | N/A | This paper strives to help readers better understand the dynamics of the changes in financial systems and the proliferation of financial products. |

**Table 1.** *Cont.*

| Item | Title | Journal | Country | Type | Scale Used | Citation | Year | Dimensions | | | | FL Definition by Author | Statistic Test Validity/Reliability | Contribution Statement | Age Scope | Sample Size | Follow-Up Time, If Applicable | Comments |
|---|---|---|---|---|---|---|---|---|---|---|---|---|---|---|---|---|---|---|
| | | | | | | | | FK | FB | FA | # | | | | | | | |
| 51 | "Edu no Planeta das Galinhas": Development process of a game about financial education for children | *CEUR Workshop Proceedings* | Brazil | Intervention | N/A | [35] | 2017 | X | X | - | 2 | Financial education serves as a cross-cutting vector that integrates the curriculum, which can help meet pedagogical objectives in education. | None | The authors propose a game to promote technology adoption in public schools to positively impact the financial education of students, families, and educators in the region. | Students, families, and educators | N/A | N/A | This article does not quantify the impact of the intervention (game development); it only explains the story per se. |
| 52 | Challenges in assessing the effectiveness of financial education programs: The Colombian case | *Cuadernos de Administracion* | Colombia | Review | N/A | [97] | 2017 | X | - | - | 1 | Financial literacy protects consumers, promotes financial inclusion, and improves financial wellbeing. | None | It demonstrates the international effectiveness of the program proposed by the authors and offers recommendations for an accurate evaluation of policies. | N/A | N/A | N/A | N/A |
| 53 | Determinants of perceived and actual knowledge of commission paid by contributors in the pension funds industry | *BRQ Business Research Quarterly* | Chile | Scale | Adapted from scale of perceived knowledge of commission paid. Adapted from scale of price consciousness. Adapted from scale about price–quality cue, price-based advertising exposure. | [98] | 2017 | - | X | - | 1 | Financial literacy has a positive effect on the real knowledge of the commissions paid by taxpayers, which makes them make better decisions. | None | The authors measure Chilean taxpayers' knowledge of commissions paid concerning the pension fund industry. | Those who plan to acquire a personal loan | 640 | N/A | The results show that price consciousness and brand credibility are positively associated with perceived and actual knowledge of commission paid by pension fund contributors. However, the results also show that financial literacy is only positively related to existing knowledge of commission paid by contributors. |

**Table 1.** *Cont.*

| Item | Title | Journal | Country | Type | Scale Used | Citation | Year | Dimensions | | | | FL Definition by Author | Statistic Test Validity/ Reliability | Contribution Statement | Age Scope | Sample Size | Follow-Up Time, If Applicable | Comments |
|---|---|---|---|---|---|---|---|---|---|---|---|---|---|---|---|---|---|---|
| | | | | | | | | FK | FB | FA | # | | | | | | | |
| 54 | Financial Literacy among Mexican high school teenagers | *International Review of Economics Education* | Mexico | Scale | Adapted from OECD scale. | [37] | 2017 | X | X | X | 3 | Financial literacy refers to a combination of knowledge, skills, attitudes, and behaviors necessary to make sound financial decisions. | None | The study provides a scale that measures the financial literacy of Mexican adolescents. | High school students ages 15 to 18 | 889 | N/A | Including both knowledge and attitudes in the regression does not significantly change the magnitude of the coefficients; these two components are uncorrelated; they measure different individual attributes. |
| 55 | Social networks and their incidence in the customer-bank relationship | *Iberian Conference on Information Systems and Technologies* | Ecuador | Intervention | N/A | [99] | 2017 | - | X | - | 1 | Financial literacy on social media is stimulated by rapid interaction and ease of access to financial products. | None | The authors contribute a social media metric to build a new content strategy to improve communication between the bank and the financial literacy program users. | Bank cus-tomers | N/A | NA | The website was analyzed using the WooRank tool. |
| 56 | Time perspective and financial health: To improve financial health, traditional financial literacy skills are not sufficient. Un-derstanding your time perspective is critical | *Time Perspective: Theory and Practice* | Brazil | Review | N/A | [100] | 2017 | X | - | - | 1 | Time perspective influences financial literacy, decision making, and financial weaknesses. | None | This paper contemplates the influence of the time perspective on financial health, hypothesizing that understanding the time perspective would be much greater than financial literacy. | N/A | N/A | N/A | This article presents definitions and an explanation of the outlook for finance in the past, present, and future time dimensions. |

**Table 1.** *Cont.*

| Item | Title | Journal | Country | Type | Scale Used | Citation | Year | Dimensions | | | | FL Definition by Author | Statistic Test Validity/ Reliability | Contribution Statement | Age Scope | Sample Size | Follow-Up Time, If Applicable | Comments |
|---|---|---|---|---|---|---|---|---|---|---|---|---|---|---|---|---|---|---|
| | | | | | | | | FK | FB | FA | # | | | | | | | |
| 57 | Mexico: Financial inclusion and literacy outlook | *International Handbook of Financial Literacy* | Mexico | Longitudinal | N/A | [81] | 2016 | X | X | - | 2 | The level of financial literacy in households can incentivize savings. | None | First, this study shows a contemporary perspective on financial inclusion in Mexico. In addition, a plan is proposed to expand financial literacy, starting with primary schools. | N/A | N/A | 66 years | N/A |
| 58 | Development of a financial literacy model for university students | *Management Research Review* | Brazil | Scale | Adapted from a survey of financial behavior. Adapted from scale of factor from the average response of two groups of multiple-choice questions. Adapted from scale of financial attitude. | [11] | 2016 | X | X | X | 3 | Financial literacy is understood as the mastery of a set of knowledge, attitudes, and behaviors and has assumed a fundamental role in allowing and enabling people to make responsible decisions. | RMSEA > 0.08 GFI, CFI, NFI, and TLI < 0.95 | This study proposes a multi-dimensional metric to determine the level of financial literacy of college students. | university students | 534 | N/A | The data were collected only in Brazil, presenting explicit peculiarities, such as an economic structure for services. Therefore, different countries should be researched using a larger sample. |
| 59 | Socioeconomic characterization and equity market knowledge of the citizens of Barranquilla, Colombia | *Lecturas de Economia* | Colombia | Scale | Elaborated by authors. | [71] | 2016 | X | X | - | 2 | The level of financial literacy determines the investment probability of individuals. | None | The authors in this paper empirically examine individuals' levels of participation and knowledge of the equity market. | Above 18 | 800 | N/A | The results show that, compared to other variables, the income range determines the level of knowledge and investment in the stock market to a great extent. |

Table 1. *Cont.*

| Item | Title | Journal | Country | Type | Scale Used | Citation | Year | Dimensions | | | | FL Definition by Author | Statistic Test Validity/ Reliability | Contribution Statement | Age Scope | Sample Size | Follow-Up Time, If Applicable | Comments |
|---|---|---|---|---|---|---|---|---|---|---|---|---|---|---|---|---|---|---|
| | | | | | | | | FK | FB | FA | # | | | | | | | |
| 60 | "Bolsa Família X" Program Financial Literacy: In search of a model for low-income women (Programa Bolsa Família X Alfabetização Financeira: Em busca de um modelo para mulheres de baixa renda) | *Espacios* | Brazil | Scale | Adapted from scale of financial attitude. Adapted from National Financial Capability Study, questions to measure financial knowledge. | [101] | 2016 | - | X | X | 2 | Financial literacy is a combination of awareness, knowledge, skills, attitudes, and behaviors required to make decisions. | Chi-Square /df > 3 GFI, CFI, NFI, and TLI < 0.95 RMSEA > 0.08 RMR > 0.05 Cronbach's alphas > 0.7 | The authors conclude that financial behavior is the dimension that most influences families' level of financial literacy. | Families | 595 | N/A | Almost unanimous presence of women in the sample (97.6%), which is justified by the federal government's decision to prioritize the granting of resources to mothers. |
| 61 | Financial Literacy among high school students in the Mexico City metropolitan area | *Trimestre Economico* | Mexico | Scale | Adapted from OECD scale. | [41] | 2016 | X | X | X | 3 | Financial literacy focuses on decisions based on financial knowledge. | None | This study contributes to the literature with a new scale that measures the financial literacy of Mexican adolescents. | High school students ages 15 to 18 | 889 | N/A | Including both knowledge and attitudes in the regression does not significantly change the magnitude of the coefficients. These two components are uncorrelated; they measure different individual attributes and thus provide additional information about students. |
| 62 | Financial literacy in university students: Characterization at the institución universitaria esumer | *Revista de Pedagogia* | Colombia | Scale | Elaborated by authors. | [45] | 2016 | X | - | - | 1 | Financial literacy helps us to understand economic processes and enables people to make wise decisions. | None | This paper designs and implements an intervention methodology that strengthens personal and social skills focused on financial literacy. | Students | 550 | N/A | N/A |

**Table 1.** *Cont.*

| Item | Title | Journal | Country | Type | Scale Used | Citation | Year | Dimensions | | | | FL Definition by Author | Statistic Test Validity/ Reliability | Contribution Statement | Age Scope | Sample Size | Follow-Up Time, If Applicable | Comments |
|---|---|---|---|---|---|---|---|---|---|---|---|---|---|---|---|---|---|---|
| | | | | | | | | FK | FB | FA | # | | | | | | | |
| 63 | Level of financial education in higher education scenarios: An empirical study on students of the economic-administrative area | *Mathematics Education* | Mexico | Scale | Banamex-UNAM test; Financial Industry Regulatory Authority (FINRA) test and National Commission for the Protection and Defense of Financial Services Users (Condusef), financial education. | [102] | 2016 | X | X | - | 2 | Financial literacy is considered a fundamental element in the decision making of personal finances. | None | This article contributes to the literature with a scale that measures the financial literacy of college teens. | College students | 115 | N/A | The results show that students have the knowledge and the habit of drawing up budgets to plan their expenses. Still, their level of financial knowledge is shallow considering the other evaluated variables. |
| 64 | Socioeconomic and financial literacy in 21st century Mexican adolescents (Alfabetización socioeconómica y financiera de adolescentes mexicanos del siglo XXI) | *Revista Electronica de Investigacion Educativa* | Mexico | Scale | Elaborated by authors. | [38] | 2016 | X | - | - | 1 | From the aspect of psychology, financial literacy relates economic knowledge with behavior. | None | The authors showed that the subjects have low financial and economic literacy levels, harming high school students. | High school students | 245 | N/A | The authors used an instrument with questions about remuneration for work, profit, interest on loans, and supply and demand. |
| 65 | The relationship of financial education and optimism in the use of credit cards (A Relação da Educação Financeira e do Otimismo no uso de Cartões de Crédito) | *Espacios* | Brazil | Scale | Elaborated by authors, financial education. Adapted from Teste de Orientação de Vida (TOV). | [103] | 2016 | X | X | - | 2 | Financial literacy helps people analyze their financial options and improves the quality of their decisions. | None | This article contributes to the current perspective of the results previously obtained in a study conducted some years ago on financial education and attitudes toward financial products. | Above 18 | 559 | N/A | N/A |

### 5. Discussion

The contribution of a designed scheme to scan and identify the fourteen variables as relevant content for each study allowed the authors to accomplish research objective 3, which may become a structured guide for future researchers. By adopting a PRISMA-based methodology [19] and the novel proposed scheme, this systematic review helped collect and process a sample of 65 articles that center on FL in LAC countries. Unlike earlier studies that adopted a narrower focus on the effects of FL [33], reviews of financial education programs [89,97], FL perspectives in the Caribbean [77], and a focus on links with health [100] or technologies [79], this review adopted a broader perspective while focusing on research published recently.

Without repeating the results reported above, the discussion section zooms in on critical elements that become obvious when summarizing the findings. First, LAC's FL research has only increased recently in the literature, especially in 2021. This result sharply contrasts with the observations of Goyal and Kumar (2021), who observed a consistent increase in studies during 2010–2021 [8], but the LAC countries do not appear in their FL review. Most research is set up in North America, Europe, and Australasia, with an apparent lower contribution of research in Africa and Asian countries, according to the study of Klapper and Lusardi (2020) [104]. The latter authors stress that, even in emerging countries, only about half of adults who use credit cards or borrow money are sufficiently financially literate. They consequently point to the risk of financially illiterate people being involved in mortgage delinquency and defaults. They emphasize the extent to which large percentages of people in such countries—and this seems applicable to LAC—lack "debt literacy", are financially fragile, and can hardly handle unexpected financial shocks as observed in the US [105].

A second observation is that most studies in this review build on the OECD definition of FL that stresses the integrated focus on knowledge, behavior, and attitudes. This definition seems accepted as a standard in research. This fact helps us to compare research findings and allows for intervention approaches that fit this FL definition and focus. The fact that up to half the studies share a comparable FL measurement instrument strengthens this position. We must stress that more studies are needed to tackle financial attitudes, an FL dimension that is often omitted.

Independent of the age focus, studies cannot neglect that FA (e.g., reflected through beliefs, self-efficacy, or perceptions) needs to be studied more extensively. The importance of this FL dimension is evident when looking at research in developed countries, such as the study of Fernandes et al. (2014) [106]. These researchers stressed the need to set up longitudinal analysis to examine how FL impacts future downstream FB. However, they noted that the effect of most available studies seems to disappear when considering the psychological traits of individuals. Lynne and Parrotta (1996) [107] underpinned financial attitude as a precursor for FK and financial behavior. The study of Qamar et al. (2016) [108] focused on the precursors to self-efficacy and attitudes when studying financial behavior.

Finally, the available research emphasizes that financial education should be a long-term undertaking. In the current study, we found few publications stressing the need for a long-term engagement with FL throughout an individual's life. Most studies seem to limit LAC FL interventions to short-term activities and to neglect what is consistently reported in the literature about the erosion of FL over time [109]. The latter cannot be overlooked considering the shifting focus of FL priorities depending on the age of citizens [110,111].

Next, the literature shows considerable differences in variables that depend on the levels of FL. The available research in other geographical areas presents a broader perspective on the downstream variables affected by FL [106]. This comment can also be linked to a final critical observation related to the present literature analysis' results. A clear description of how changes in FL are linked to key financial decisions or other outcome variables is often absent. These points lack a clear theoretical framework to explain how or why interventions might have an expected impact, a shortcoming already mentioned in earlier FL research [112]. Examples of such theoretical groundings are scarce but inspiring. For

instance, Amagir et al. (2017) built on the self-determination theory of motivation to predict the impact of FL interventions [113]. Others work with the human capital theory [114], the theory of planned behavior (TPB), the theory of consumer socialization (TCS), or the social learning theory [8].

Theoretical frameworks could help clarify the dominant FL conceptualization's relationship between attitudes, knowledge, and behavior. They could be enriched with variables that might interact with changes in or levels of FL, such as gender or cognitive capacities. The latter variables are critical considering the consistent results pointing to gender differences in FL [22,24,25], highlighting the difficulties women experience in accessing and using technologies. Potrich et al. (2018), in their research, found that men have a higher level of FK than women, which is why they recommend working on financial inclusion programs focused on single and low-income women [17]. On the other hand, in FB, gender is significantly related to other variables such as income and marital status. As a result, women are less likely to be taxpayers. They are less involved in finance than men [28].

This study developed broad conceptualizations regarding FL since they can vary and include new terms depending on the research context. FL has been linked to topics such as sustainability [86], artificial intelligence [28], and health [29], which shows that there is a flexible interconnection with other areas of knowledge. This relation between literacy and other topics may seem alien to a financial approach, but it has allowed the development of programs and policies that promote financial inclusion. This was done in countries such as Mexico, given their low levels of FL in secondary education [37]. At a global level, there are also entities such as S&P Global FinLit that contribute to the design of policies intending to improve financial wellbeing and offer tools to measure the level of FL to identify the weaknesses and strengths of the population [115]. The importance of measurement tools known as scales or surveys lies in the fact that they allow us to see the situation and thus establish policies according to the needs of each country [46], such as those aimed at reducing gender or technological gaps [81].

## 6. Conclusions

FL has gained global relevance and importance over time, mainly due to the expansion of financial markets, the wider variety of financial products available for users, and the financial needs linked to the growing involvement of citizens in microlevel economic processes. This fact has introduced the need to include an adequate level of FL into education. Economic changes also affect countries in the LAC region. However, research focusing on mapping the FL of citizens from these countries is scarce. The present systematic review contributes to filling this gap in the literature. As a result, the assessment of the status of FL among LAC countries mirrors worldwide trends in that it shows deficiencies. The key finding is the low number of FL studies. They lack a clear theoretical framework and consistently aim to develop all FL dimensions, focus on short- and long-term objectives, and consider critical background variables in citizens.

The current research offers three crucial outputs: a picture of the state of FL research in LAC showing its progress and interest compared to the worldwide benchmark; a transparent process and guide to make and compare future evaluations of the FL research in the region; and a diagnosis showing the shortcomings concerning FL research and what can be done to enhance the results.

The needed improvements in FL research in LAC involve increasing the quantity of FL research production in the region and the quality of the publications. The quality factor implies, for example, a need to analyze FL with an enhanced conceptual framework, include the often-forgotten financial attitude dimension of FL, approach the FL endogeneity bias increasingly studied worldwide, and link FL changes to key financial decisions (house, pension plan purchasing, etc.).

LAC needs to counter the lack of debt literacy in the region and a financial fragility that can hardly handle unexpected economic shocks, as observed in developed countries. There is an increasing need for studies emphasizing long-term engagement since there are

consistent reports in the literature about the erosion of FL over time. These factors should all be considered in our efforts to design better solutions that attend to all related needs.

This work contributes to a sample comparison of thirty LAC countries through a formal methodology named PRISMA. The contribution of a scheme designed for scanning and identifying the fourteen variables as relevant content may represent a structured guide and constitute a key output for future researchers.

The main limitation of our study is that we may have missed studies published in reports, dissertations, and books. Additionally, the focus on journals incorporated in Scopus may have limited our search. Future research could expand the search and start from an a priori FL framework to consider how regional analysis differs from generic models that look at the need for FL. Economic structures can interact with and amplify regional differences depending on specific cultural and socio-economic settings.

**Supplementary Materials:** The following supporting information can be downloaded at: https://www.mdpi.com/article/10.3390/su14073814/s1, File S1: The PRISMA checklist 2020.

**Author Contributions:** Conceptualization, S.M.M.P.; methodology, S.M.M.P., S.G.Z.Z. and M.J.Z.F.; software, K.M.C.G.; validation, S.M.M.P. and M.V.; formal analysis, S.G.Z.Z., M.J.Z.F. and M.V.; investigation, S.G.Z.Z. and M.J.Z.F.; resources, S.M.M.P.; data curation, S.G.Z.Z. and M.J.Z.F.; writing— original draft preparation, S.G.Z.Z. and M.J.Z.F.; writing—review and editing, S.M.M.P., S.G.Z.Z., M.J.Z.F. and M.V.; visualization, S.G.Z.Z. and M.J.Z.F.; supervision, S.M.M.P.; project administration, S.M.M.P., P.E. and K.CH.; funding acquisition, S.M.M.P. All authors have read and agreed to the published version of the manuscript.

**Funding:** This research received no external funding.

**Institutional Review Board Statement:** Not applicable.

**Informed Consent Statement:** Not applicable.

**Data Availability Statement:** Not applicable.

**Acknowledgments:** This work was carried out thanks to Escuela Superior Politécnica del Litoral (ESPOL).

**Conflicts of Interest:** The authors declare no conflict of interest.

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
