# Peer review of "A Systematic Review of Financial Literacy Research in Latin America and The Caribbean"

_sustainability, doi:10.3390/su14073814_

Round 1
Reviewer 1 Report
The FL research production gave us over 4,500 manuscripts worldwide, but only 65 articles were finally related to our LATAM countries' analysis scope. Being the first FL review analyzing the member countries of LATAM.
Author Response
Dear Reviewer:
Please see the attachment:
- The letter response for all reviewers
- The "TRACK" manuscript file with all visible changes.
- The new manuscript with the accepted changes for your review with the letter response.
Thank you.

Reviewer 2 Report
Dear Authors,
Thank you for the opportunity to review your manuscript titled "How about Financial Literacy research on Latin America and The Caribbean? A systematic review". I have some minor suggestions for you to consider.
- The title - could you modify it? Perhaps, "A systematic review of financial literacy research in Latin America and The Caribbean."
- The introduction - "How much an individual accumulates wealth, debt, risk tolerance, and retirement preparedness." does not work as a stand alone sentence. Perhaps, add "For instance, how much...."
- The introduction - "Researchers publish about financial literacy." Oddly placed sentence. Please expand or remove.
- In the introduction you mention financial literacy a lot but haven't defined it. You need to include a definition earlier in your paper.
- Conceptual framework - assumptions are made about individuals who are financial literate. For example, "Individuals will build their skills and abilities, leading them to control their finances better." There are financially literate individuals living on low incomes and there is research published about this. Try to avoid deficit assumptions about individuals who are living on low incomes.
-
Throughout your paper you use FL and sometimes Financial literacy. Be consistent.
-
Not sure about your referencing style. Please double check to ensure consistency. - You mention the Table but do not explicitly state "see Table x" in the appendix.
- Conclusion - "c" in missing from the word "economic" in this section.
- Conclusion - "The findings show the financial literacy state in LATAM nowadays by using the process and contributions related to this work to make future evolution evaluations of the FL research in the region. " I have no idea what this means. Please revise and make clearer.
- Conclusion - "A needed focus is not only on the quantity of research production, but in an enhanced conceptual framework." Not sure what this means either. Please clarify.
- Conclusion - "Also, studies to tackle financial attitudes, the FL dimension, which is often omitted, and to battle the lack of debt literacy characterizing the region with a financial fragility that can hardly handle unexpected economic shocks as observed in developed countries." Not sure what you are trying to say here. Please revise.
- I think the paper could be strengthened by adding what the implications are from this research.
Author Response
Dear Reviewer:
Please see the attachment:
- The letter response for all reviewers
- The "TRACK CHANGES" manuscript file with all visible changes.
- The new manuscript with the accepted changes for your review with the letter response.
Thank you.

Reviewer 3 Report
A well-written text. It is a work of literature review, for a region of the terrestrial globe. It is delimited by methodology. There is nothing to add.
Author Response

(The authors gave the same response as above.)

Reviewer 4 Report
This study reviews financial literacy literature in the LATAM region. The review mostly focuses on the measurement issues of financial literacy. However, the lack of a well-directed review on how FL affects economic and financial behavior limits its potential.
- Lines 19-20 – I did not find any reference in the text that shows the comparison.
- Lines 35-36 need reference
- Motivation and contribution of this study is not clear
- Line 88- Do ref 8 and 10 compare three components of FL?
- Authors need to review studies that directly compares among the components of financial literacy i.e., FK, FA, and FB to better understand how these components are related.
- Line 108 – how about terms like financial knowledge, financial behavior, etc.
- Line 110 – why did the authors restrict the search within 2016-2022 timeframe?
- Authors could expand their search options, find more studies in LATAM, and remove some overlapping studies, if needed.
- Endogeneity is a serious concern in FL studies. How did the studies in LATAM region consider this issue?
- Authors seem to limit the review on the definitions and components of FL. Authors could do better if they reviewed how those studies associated FL with economic, financial, and other behavioral aspects.
- There is no direction for future research based on what the authors have found in the review
Author Response

(The authors gave the same response as above.)

Round 2
Reviewer 4 Report
The authors have substantially improved the manuscript and most of my comments have been responded satisfactorily.
However, I would request the authors for copyediting of the manuscript by a native English speaker. There are grammatical mistakes in several places. The manuscript lacks clarity and flow as well.
Author Response

(The authors gave the same response as above.)
